# Daylight Photoactive TiO_2_ Sol-Gel Nanoparticles: Sustainable Environmental Contribution

**DOI:** 10.3390/ma16072731

**Published:** 2023-03-29

**Authors:** Daniel Alves Barcelos, Maria Clara Gonçalves

**Affiliations:** 1Departamento de Engenharia Química, Instituto Superior Técnico, Universidade de Lisboa, Av. Rovisco Pais, 1049-001 Lisboa, Portugal; danielbarcelos@tecnico.ulisboa.pt; 2Centro de Química Estrutural, Av. Rovisco Pais, 1049-001 Lisboa, Portugal; 3Centro de Ciências e Tecnologias Nucleares, Instituto Superior Técnico, Universidade de Lisboa, 2695-066 Bobadela, Portugal

**Keywords:** sol–gel, doped titania, composite titania, amorphous titania

## Abstract

Visible-light-photoactive titania micro- or nanoparticles excel in a wide range of industrial areas, particularly in environmental remediation. The sol–gel methodology is one pivotal technique which has been successfully used to synthesize either crystalline and amorphous TiO_2_ micro- and nanoparticles due to its outstanding chemical simplicity and versatility, along with the green chemistry approach. This short review aims to collect and discuss the most recent developments in visible-light-photoactive titania-based nanoparticles in the environmental remediation area. Titania co-doping, titania composite design, and, recently, amorphous networks have been the most used strategies to address this goal. Finally, a prediction regarding the future of these fields is given.

## 1. Introduction

Since its commercial production in the early 20th century, titania (titanium dioxide, TiO_2_) has been the most widely used white pigment (used as pigment-grade micro- or nanoparticles (NPs)) [1] due to its optical transparency, brightness, and extremely high refractive index (anatase, *n*_500 nm_ = 2.6, *n*_633 nm_ = 2.48–2.49; rutile, *n*_500 nm_ = 2.74, *n*_633 nm_ = 2.57–2.62; brookite, *n*_500 nm_ = 2.8, *n*_633 nm_ = 2.61–2.67 [2]), for which it is surpassed only by a few other materials such as GaP (*n*_1.6 mm_ = 3.05), Si (*n*_1.6 mm_ = 3.47), or Ge (*n*_1.6 mm_ = 4.24) [3]. Formulations such as paints and coatings, tattoo pigments, styptic pencils, or paper are just some of the pigment-grade illustrations [1,4,5]. In architecture, civil engineering, and the automotive industry, titania pigments are used in coatings such as paints, stains, lacquers, clear coats, or primers, where they bring longevity to the paint and protection to the substrate [6,7,8,9,10,11,12,13].

Moreover, titania is the only ceramic product which is Food and Drug Administration (FDA)-approved as a food additive (E171) [14], and is commonly used to enhance the white color of dairy products and candies. Furthermore, its value has been extended as a flavor enhancer in a variety of nonwhite foods, including dried vegetables, nuts, seeds, soups, and mustard, along with beer and wine.

Titania pigments add brightness to cosmetics (shampoos, balms, lotions, creams, soaps, lipsticks, toothpaste, and hair dyes), ointments, and some medications (also FDA-approved [15]). In sunscreens, and at the nanoscale, titania NPs (usually coated with silica and/or aluminum hydroxide and often combined with chemical ultraviolet filters (UV-filters) such as avobenzone) do not reflect visible light, but instead do absorb UV radiation (due to the particle size), enabling a transparent protecting barrier from the sun’s UV rays (FDA-approved [16]). According to the Skin Cancer Foundation, the use of sunscreens containing titania helps prevent the occurrence of skin cancer [17]. In sunscreens developed for infants or people with sensitive skin, titania is the selected UV blocker as it is less likely to cause skin irritation than any chemical UV absorber ingredient.

Being a photoresponsive material, titania particles protect plastics, adhesives, and rubbers from UV degradation, increasing their durability. Titania micro- and NPs are extensively used in polyvinyl chloride and engineered resins owing to their broad compatibility, ease of dispersion, and ease of processing, in various polymer types such as ABS, acrylics, PET/PBT, PVC, polyamide, polycarbonate, polyethylene/polypropylene, and polystyrene [18]. When used in food contact materials, titania’s opacity to visible and UV light protects food, beverages, supplements, and pharmaceuticals from premature degradation, enhancing the product’s longevity.

Catalysis is the second great titania particle application. Titania micro- and NPs and mesoporous titania [14] are widely used as support materials in catalyst applications. Major uses include the automotive industry, in the reduction in harmful exhaust gas emissions or nitrous oxides in power stations. However, when the photocatalytic splitting of water on a TiO_2_ electrode under UV was discovered in 1972, TiO_2_ came into its own [19]. This has led to reams of research into TiO_2_ UV photocatalysis, owing to its usefulness in areas such as photovoltaics, photocatalysis, and electrochemistry. Water splitting, self-cleaning, antifogging, sterilization, disinfection, the prevention of stains, lithography, the photodegradation of organic pollutants or the catalytic oxidation of carbon monoxide, metal corrosion prevention, dye-sensitized solar cells, and smart materials, are among the new actions able to be performed by TiO_2_ based-NPs due to its strong oxidizing power when bombarded with UV radiation, which has led to its widespread use throughout a number of industries [20,21,22,23,24,25,26,27].

Moreover, TiO_2_ is highly hydrophilic with a water contact angle of 0° [28]. In the architecture, civil engineering, automotive, and aerospace industries, titania brings antifogging performance and self-cleaning properties to coatings, paints, floors, and glass windows. In cities, when introduced into outdoor building materials (usually as TiO_2_ NPs) such as paving stones, it appears that titania helps lower concentrations of toxins such as volatile organic compounds and nitrogen oxides in the air [29].

In environmental protection, TiO_2_ plays an important role in water treatment through advanced oxidation processes (AOPs) due to its photocatalytic activity. AOPs are an important alternative used to transform dangerous substances into inert compounds through full mineralization. The use of titania for this purpose started 3 decades ago but there is still room for improvement by addressing titania visible-light sensitivity and the annulment of electron/hole recombination [30].

### 1.1. Titania in the Earth’s Crust

Because of its great reactivity towards oxygen, metallic-state titanium cannot be found in nature, and instead makes up four recognizable oxides—titanium monoxide (TiO), dititanium trioxide (Ti_2_O_3_), titanium dioxide (TiO_2_), and titanium trioxide (TiO_3_). The elemental abundance of titanium in the earth’s crust is estimated to be about 0.64 weight percent (or 1.07 as titania [31]), a far distance from the 75 weight percent of silica. Note that titanium is reported in the chemical analyses of rocks and minerals as TiO_2_.

The natural occurring TiO_2_ polymorphs—anatase (tetragonal), rutile (tetragonal), brookite (rhombohedral), and TiO_2_(B) (monoclinic) [1,27,32]— all exhibit a common structure. Rutile, the thermodynamic stable phase at ambient temperature and pressure [33,34], is present in diverse geological environments as the alteration products of metamorphic and igneous rocks. Anatase (metastable at ambient conditions) scarcely appears in the veins of igneous and metamorphic rocks, being present in the detrital deposits of Minas Gerais and Bahia, Brazil, in the Sanarka region of the Urals, and in European Alps. Brookite is a relatively uncommon mineral found as an accessory in several igneous and metamorphic rocks including granites, syenites, gneisses, and schists, and may be formed in hydrothermal veins. It is resistant enough to weathering to survive as a heavy mineral in sediments. Ilmenite (FeTiO_3_), a common accessory mineral in a wide variety of rocks, is the main geological titania source. Natural ilmenites show an average TiO_2_ content of 52.5 weight percent (see [34]); small variations are due to the presence of intergrowths with hematite (Fe_2_O_3_) and magnetite (Fe_3_O_4_), to the solid solution with hematite (low values), and to the leaching of iron during weathering. When the TiO_2_ content of altered ilmenite exceeds about 70 weight percent it is commonly referred to as leucoxene (TiO_2_.xFeO.yH_2_O). Leucoxene, which is mined from deposits located throughout the world, is an alteration product and mixture of Fe-Ti oxides, including titanite, perovskite, and titanian magnetite, but especially ilmenite.

Titania polymorphs find different uses. The rutile phase is usually used in high-grade, corrosion-protective white coating and paint, or in plastics, rubber, leather, sunblock lotion, and paper due to its high refractive index. The anatase phase has excellent optical and pigment properties due to its electronic structure and it is used as an optical coating and photocatalyst. The applications of brookite are limited because it is uncommon. Raw material suppliers are the first stage of the value chain for industrial titania production. Table 1 shows a short summary of titania polymorphs’ properties.

Besides naturally occurring titania, titania originating from human activities is also present in Earth’s crust where it could be a potential environmental problem. Titania NPs are one of the most produced nanomaterials worldwide, being present in many consumer products. After use, a large amount of TiO_2_ NPs ends up in agricultural soils through the application of sewage sludge. A prerequisite for risk assessment is to be able to distinguish this released TiO_2_ from TiO_2_ natural background, and, therefore, detect anthropogenic inputs.

### 1.2. Titania Polymorphs

*Rutile* and *anatase*, the most studied and used industrially titania polymorphs (Figure 1), share the same unit cell structure, a TiO_6_ octahedron (with a central Ti^4+^ ion in the O^2−^ octahedron cage). In *rutile,* a slight orthorhombic distortion is present in the unit cell, while in *anatase,* a significantly distortion is observed (with the symmetry being lower, as two of the titanium–oxygen bonds are much longer than the other four bonds and the O-Ti-O bond angles deviate more from 90° than in rutile). Further, rutile forms a linear chain structure where each octahedron is in contact with 10 other octahedrons (2 sharing edge oxygen pairs and 8 sharing corner oxygen atoms), while anatase forms a zigzag chain with a screw axis (where each octahedron shares 4 corners and 4 edges with other octahedrons). These differences in lattice structures cause different mass densities and electronic band structures between the two TiO_2_ polymorphs.

Brookite has an orthorhombic cell. Though the interatomic distances and O-Ti-O bond angles are similar to those of rutile and anatase, there are six different Ti-O bonds ranging from 1.87 to 2.04 Å in length. There are also twelve different O-Ti-O bond angles ranging from 77° to 105°. Surprisingly, brookite and rutile are structurally similar because both phases are formed by straight polyhedron chains linked through three different corners of the unit cell but linked polyhedron chains occur through cis bridges in brookite.

At elevated temperatures, anatase, brookite, and TiO_2_(B) will phase-transform: anatase and brookite → rutile, anatase → brookite → rutile, brookite, and TiO_2_(B) → anatase → rutile [1,27,38] (Figure 2). Notwithstanding, anatase may also transform into an amorphous network (rather than to rutile) under particular conditions—high-energy milling operations [39] or crystallite refinement [40] are just two examples.

Despite rutile being the thermodynamically stable phase at ambient conditions, the first to crystallize in many synthetic routes is anatase, thanks to structural characteristics such as (i) its more flexible assembly of the 4-edge-sharing TiO_6_ octahedron, (ii) the presence of impurities, structural defects, and (iii) of crystal growth conditions. Moreover, at the nanoscale and for equal NP size, anatase is the most thermodynamically stable at sizes less than 11 nm, brookite is most stable for crystal sizes between 11 and 35 nm, and rutile is most stable at sizes greater than 35 nm (Figure 3).

Regarding amorphous TiO_2_ nanoparticles (a-TiO_2_ NPs), a structural contribution was published recently by the authors [43]. Slightly distorted anatase-like crystalline core (with a dimension of approximately two unit cells) enveloped by an outer distorted shell (of approximately 2–4 atomic layers thick, of octahedral-like structure and Z_Ti-O_ = 5.3 and Z_O-Ti_ = 1.94 [44,45,46]) seems to be the model that best fits the experimental results. While the NP’s core structure is independent of the NP’s size, the NP’s size is a deciding factor in the structure of the shell. Many defects in the structure thereof such as under-coordinated sites such as TiO_3_, TiO_4_, or TiO_5_ and dangling bonds can be seen in the shell [47]. These have a large part to play in TiO_2_ NPs’ properties. Using a computer simulation, we can confirm the presence of a distorted octahedral core structure (with Z_Ti-O_~6.0 and Z_O-Ti_~3.0), while the shell tends to be more porous and feature more defects (with Z_Ti-O_ ≠ 6.0 and Z_O-Ti_ ≠ 3.0) [48,49], driven to a density value (a-TiO_2_ NPs 3.74 g/cm^3^) lower than the bulk anatase density (3.90 g/cm^3^). With reference to the electronic band gap in crystalline titania, it is well established, both theoretically and experimentally. Nevertheless, the understanding of the structural and electronic properties of amorphous titania phases (a-TiO_2_) is closer than ever, thanks to ab initio calculations for a-TiO_2_, and predicated upon the shared features of its form under the electronic band of its amorphous and TiO_2_ crystalline phases [43].

### 1.3. Titania Synthesis

Industrial titania pigments (pigments grade) are manufactured from a variety of ores containing natural minerals such as ilmenite, rutile, anatase, and leucoxene. Mining and chemical refining are the two industrial steps for raw material preparation. The vapor phase and liquid phase are the main chemical refining routes. Today, more sustainable methodologies are on the industrial agenda.

Synthetic titania micro- or NPs are, in the laboratory or industrially, produced through atomic layer deposition (ALD) [50,51,52], extraction pyrolytic [53,54,55], hydrothermal [56,57,58] and solvothermal [59,60,61], or sol–gel [62,63,64] methodologies. The authors’ group developed a novel, room-temperature, base-catalyzed, sol–gel protocol to synthesize amorphous TiO_2_ NPs [65], based on the classical Stöber method (Figure 4). The sol–gel chemistry constitutes an environmentally friendly synthesis route, with deep roots in the so-called green or soft chemistry.

Hydrothermal or solvothermal processes remain a key technique for producing nanomaterials, for which countless scientific publications have been published and a huge number of products have been produced. From the perspective of morphology, a wide variety of morphological features have been fabricated, namely nanosphere, nanotube, nanorod, nanowire, nanobelt, and nanoplate. The synthesis of monodispersed nanoparticles with complete control over morphology and size distribution, in addition to their chemical homogeneity and greatest dispersibility, is one of hydrothermal technology’s major benefits for processing nanomaterials [66].

The sol–gel method has been widely used to synthesize TiO_2_ films and coatings usually through an acid-catalyzed route [67]. The TiO_2_ sol–gel process involves the hydrolysis of titanium precursors, primarily alkoxides and chlorides, followed by condensation. These reactions result in the formation of Ti-O-Ti 3D networks, the yield of which is determined by the water/alkoxide ratio, the presence of acid/base catalysts, the solvent type, gel aging, the operating temperature, and the mixing technique. Furthermore, hydrolysis and condensation are competitive, and it is preferable to separate and temper these two steps to gain better control over the evolution of the porosity and morphology of TiO_2_ films. Several approaches were used to achieve this goal. One of them is the modification of alkoxides by complexation with coordinating agents that are less easily hydrolyzed, i.e., more strongly bonded to the metal, such as carboxylates or b-diketonates [68]. In most cases, sol–gel reactions are followed by a thermal treatment to remove the organic component (~40–80 °C) or to crystallize either anatase or rutile TiO_2_ (450–1000 °C). Calcination will inevitably result in a decrease in surface area and mesoporosity (due to sintering and crystal growth), a loss of surface hydroxyl groups, and even phase and morphology transformation [68].

The observed sensitivity to sol–gel reaction conditions suggests that it should ultimately be possible to obtain good control over titania NP synthesis.

### 1.4. Titania Photocatalytic Activity

When a photon (hv) of energy equal to or higher than the semiconductor band gap (3.23 eV for anatase; 3.0 eV for rutile) is diffused by the material, a photocatalytic reaction is triggered. A ‘hole’ in the VB appears when the photoexcited electron is promoted from the (filled) valence band (VB) to the (empty) conduction band (CB). When this electron-hole pair appears, there are three possible outcomes that will determine the fate:(i).Recombination path: This is the most common result, in which, without any chemical effect, ~90 % of the *e^−^*-*h^+^* pairs quickly recombine owing to a rapid dissipation as heat [21,22].(ii).Trapping path: This takes place in around 8% of events and is where a charge or both charges are transported to defects in the crystalline network. This makes for a slightly weaker reductant or oxidant, but on the other hand, the charge separation is preserved, perhaps allowing further photocatalysis reactions [21,22].(iii).Reactional path: Accounting for only 1–2% of observed events, this is where photogenerated charges lead to reactions with surface-absorbed molecules. Radicals, that is with only one unpaired electron, form here due to the upset caused by the lone electron present in each redox reaction [21,22]. One or both charge carriers (*e^−^*, *h^+^*), after migrating to the material surface, react with surface-adsorbed molecules. As each redox reaction involves only one electron, the even number of valence electrons usually found in molecules will be upset, and radicals (with one unpaired electron) form.

The strong oxidant power of *h*^+^, which is able to strip one electron from an absorbed molecule, promotes oxidizing reactions. Organic radicals form in direct reactions, while hydroxyl and superoxide anion radicals form in indirect ones:h++H2O→O•H(ad)+H+
h++O2−(ad)→2O•(ad)

^•^OH and ^•^O are very powerful oxidants. They oxidize most molecules they contact. Reducing reactions are carried out by photoexcited *e^−^*. In this case, adsorbed organic molecules are reduced (by taking *e^−^*), and O_2_ (or H_2_O) form ^•^O_2_^−^:$$e^{-}+O_{2}\to O_{2}^{-}(ad)$$
$$O_{2}^{-}(ad)+H^{+}\to HO_{2}^{•}(ad)$$

The superoxide radical anion, reacting with water, oxygen, and additional *e^−^*, thus produces a shower of reactive oxygen species (ROS), like ^•^HO_2_^−^, H_2_O_2_ and ^•^OH^−^, which are of great importance in sterilization and antimicrobial outcomes. In environmental remediation these radicals will react with pollutants, leading in many cases to mineralization, the process which decomposes the pollutant into CO_2_ and H_2_O.

In environmental remediation, these radicals will react with the pollutants, leading in many cases to mineralization, which is the process that decomposes the pollutant into CO_2_ and H_2_O. A scheme of this particular application is shown in Figure 5.

## 2. Strategies to Enhance Daylight Titania Nanoparticles’ Photoactivity

The increased photoactivity of TiO_2_ NPs under visible light is a hot topic in materials science, aiming at indoor/sunlight TiO_2_ applications. Three main strategies will be visited—titania composites, titania doping (with a metal or nonmetal element), and amorphous titania.

### 2.1. Titania Composite

The simplest way to optimize titania photocatalyst activity is by preventing the recombination of photogenerated electron–hole pairs. This can be achieved in composite structures by mixing the titania semiconductor with a conductor material (i.e., a material with high electrical properties, namely higher electron mobility). By preventing the recombination of photogenerated electron–hole pairs (due to the π–π interactions of graphene, as an example), the charge transfer rate increases, and thus does the photocurrent. Graphene (with an electron mobility higher than 2000 cm^2^/V and a work function of 4.5 eV) emerges as an excellent candidate for this purpose. The academic community has focused great attention on this approach by combining titania with carbon-based materials such as graphene (GO), multiwalled carbon nanotubes, or metal–organic frameworks (MOF), as examples.

Zhang et al. produced TiO_2_ nanofibers (with sol–gel electrospinning) and composed them with GO (from 0 to 5 wt%). The TiO_2_-GO composite showed enhanced photocatalytic properties and optical absorption in the visible range. The optimum GO concentration (aiming at photocatalytic efficiency) was 4 wt% [69]. Another example of a TiO_2_-carbon composite is based on polymeric graphitic carbon nitride (g-C_3_N_4_). g-C_3_N_4_ exhibits high thermal and chemical stability and a narrow band gap. Zada et al. additionally proposed an Au heterojunction in the composite—Au-(TiO_2_/g-C_3_N_4_) (Au from 0.5 to 4 wt%, TiO_2_ from 2 to 8 wt%) [70]. The authors were able to promote the dichlorophenol (DCP) and bisphenol photodegradation under visible light (46% and 37%, respectively). A related work was published by Porcu et al. [71]. Here, a TiO_2_-phenyl-carbon nitride composite was (hydrothermally) synthesized and tested as a photocatalyst for rhodamine B and methylene blue (and compared with a g-C_3_N_4_/TiO_2_ system). The composite system successfully degraded 98 % of rhodamine B and 88% of methylene blue, but after four cycles, its efficiency went down to 76% in methylene blue. These results reinforce the idea that phenyl-carbon nitride opens a bright future for such hybrid composites.

Koli et al. [72] produced a composite TiO_2_-multiwalled carbon nanotubes. The carbon nanotubes were mixed with titanium (IV) isopropoxide and left for 2 h at 60 °C. The product was dried at 110 °C and calcinated at 450 °C for 4 h. Concentrations from 0.1 to 0.5 wt.% of carbon nanotubes were used. The composites were tested with methyl orange dye under UV and solar light. The best results came from 0.5 wt.% of carbon nanotubes under UV (it requires 4× time to reach the same result under solar light, i.e., 30 min. under UV and 120 min under solar stimuli).

A sea-urchin-like composite was fabricated with Fe_3_O_4_ (core), TiO_2_ (shell), and Ag coating, through sol–gel and hydrothermal methods (Zhao et al. [73]). The authors further addressed the recovery and reuse of the material by adding magnetic properties, in addition to the visible-light activation. The composite exhibited a band gap of 1.5 eV (much lower than the pristine TiO_2_~3.2 eV). Satisfactory ampicillin degradation (92%) and good antibiosis response were observed [73]. Liu et al. [74] used polydopamine to synthesize a composite of TiO_2_ and glutaraldehyde (through the sol–gel method). The use of dopamine is quite interesting due to its surface versatility (enabling polymerization with organic and inorganic molecules), besides its multiple active sites to interact with organic pollutants. Similarly, the use of glutaraldehyde allows the recovering and reuse of the composite NPs. The composites’ photonic band gap are 2.28 eV (TiO_2_/PDA NPs) and 2.25 eV (TiO_2_/PDA/glutaraldehyde coating) proving the capacity of photocatalysis under visible and UV light. Geosmin and fluorene were used to evaluate the composite degradation capacity. TiO_2_/PDA NPs exhibited a photodegradation capacity up to 90% for geosmin and 99% for fluorene; TiO_2_/PDA/glutaraldehyde NPs exhibited 91.5% for geosmin and 99% for fluorene [73]. Cai et al. [75] used polystyrene (PS) spheres as a sacrificial template to build the following double-shell nanocomposites—TiO_2_-CeO_2_, TiO_2_-CeO_2_/Au, and TiO_2_-Au-CeO_2_, through a sol–gel approach. Cerium oxide was selected due to its thermal stability and oxygen storage capacity, promising for advanced oxidative process, as well as gold due to its electron conductivity and high passivation. The best structure configuration appears to be the one with Au in the middle core, possibly due to the synergistic effects between CeO_2_ and Au [75].

Jingsheng Cai et al. [76] produced Bi_2_MoO_6_ (nanosheet)-TiO_2_ nanotube arrays doped with Au NPs (via the solvothermal method) and tested the photocatalytic capacity by degrading methylene blue and benzene series compounds (phenol and bisphenol A). The composite showed a wide wavelength adsorption from 200 to 800 nm, and excellent photodegradation activity against the targets used. The plasmon resonance effect from Au and electronic capture from Bi_2_MoO_6_ contribute to a longer lifetime in the pairs (*e*^−^ _CB_/*h*^+^
_VB_) and prevent their recombination [76]. Perween and Ranjan [77] sol–gel-synthesized ZnTiO_3_ nanopowders. A surfactant (cetyltrimethyl ammonium bromide (CTAB)) and electrospinning route were used. A phenol solution was used as pollutant model in the visible-light range (420 nm and 480 nm). The best photocatalytic activity was reached using nanopowder electrospin synthesis [77].

Metal–organic frameworks (MOFs) are 3D porous materials with huge surface area and large surface cavities, formed by metals bonded to multidentate organic molecules. Lanjie Li et al. used a metalloporphyrin-MOF (PCN-222 (Mn)) and produced a composite material with titania and H_3_PW_12_O_40_ (PCN-222-PW_12_/TiO_2_), where PCN-222 varied from 2% to 10 wt%. The pollutants tested were rhodamine B and ofloxacin [78]. The composite was able to extend the absorption of light from 200 nm to 800 nm, like bulk PCN-222. The best degradation results were obtained with 5 wt% PCN-222, these being 98.5% and 94.8% for rhodamine B and ofloxacin, respectively.

### 2.2. Titania Doping

Another commonplace strategy for reducing the rutile or anatase band gap, carried out by metal or nonmetal elements, is adding a visible-light sensitization dopant in situ. In substitutional doping with other *3-d* transition metal atoms, a lower energy level of metallic *d* states is introduced in the titania band gap, lowering its CB. When doping with anions is the choice, the nonbonding p_π_ state of O shallows or mixes with the band states of the dopant anion, uplifting the VB.

Nonmetal nitrogen (in a substitutional solid solution) showed the most promise in presenting more visible-light sensitizer opportunities for titania, where dominant transitions at the absorption edge were identified with those from N 2*_px_* to Ti d*_xy_*, instead of those from O 2*_pπ_* to Ti d*_xy_* [40,42]. Recently, Herrera et al. doped sol–gel TiO_2_ NPs with Cu (1–2 mol%), aiming at the daylight purification/disinfection of effluents. An amount of 2 mol % of Cu was able to narrow the TiO_2_ band gap up to 2.86 eV. Under visible light, 41% of diclofenac was degraded after 7 h of exposure, and there was a decay of 42% of the chemical oxygen demand (within the same time) in a real effluent sample treatment [79]. Li et al. [80] developed a method to obtain N-doped titania sol–gel NPs. Essentially, they used tetrabutyl-orthotitanate and ethylenediamine (from a 0 to 3 ratio with the solution), stirred the solutions for 4 h, aged them for 3 days, dried them at 60 °C for another 5 days, and finally calcined for 1–3 h at 300–600 °C. The photocatalytic activity was detected with methyl orange. The best heat treatment was 500 °C and the best N:Ti ratio was 1:1. Tobaldi et al. [81] studied the N doping of sol–gel TiO_2_ NPs (in acidic and basic conditions). Under acidic conditions, NO_3_ + NH_3_ and HNO_3_ were used in a Ti:HNO_3_ 5:1 molar ratio; under basic conditions, NH_4_OH was used in a Ti:NH_3_ 2.5:1 molar ratio. In both cases, the synthesized powder was dried at 75 °C overnight, followed by heat treatments from 450 to 800 °C. The titania precursor was titanium (IV) isopropoxide. The photocatalytic activity was measured in terms of NO_x_ degradation. The best catalyst activity (a degradation of 34% NO_x_; the results remained constant after three runs) was obtained with TiO_2_ NPs synthesized via the acidic route and heat-treated at 450 °C for 20 min. Under white light, the performance decreased to 19% for the acidic-route product compared with 21.5% for the basic one (both treated at 450 °C). The results were superior to the commercial photocatalyst P25 (11–13% degradation in the same conditions) [81].

Metal atoms with strong UV transitions have also been tested as light sensitizers. Li et al. [82] synthetized titania sol–gel NPs codoped with molybdenum, antimony, and sulfur—S-TiO_2_, Mo-S-TiO_2_, Sb-S-TiO_2_, and Mo-Sb-S-TiO_2_. Tetrabutyl titanate, (NH_4_)_6_Mo_7_O_24_·4H_2_O, SbCl_3_, and thiourea were the precursors, the Mo:Ti and Sb:Ti molar ratios were 0.06 and 0.02, and the heat treatment was 80 °C for 12 h, followed by 3 h with a constant heat flow of 5 °C. The target pollutant was methylene blue. After 2 h under visible photoactivation, the methylene blue degradation efficiency was Mo-Sb-S-TiO_2_ > Mo-S-TiO_2_ > Sb-S-TiO_2_ > S-TiO_2_. Factors such as large specific area, high crystallinity, crystalline size, porous structure, and intense absorption in the visible light explained the efficiency of the doped TiO_2_ NPs [82]. Yu et al. [83] synthesized titania sol–gel NPs doped with V. titanium butoxide and vanadyl acetylacetonate were the precursors, and the photocatalytic activity was proved in the methylene blue degradation. The TiO_2_ sol–gel NPs were dried for 4 h at 70 °C, and then calcinated at 400–600 °C for 4 h. The V:Ti ratio ranged from 0.15% to 0.45 mol%. From the photodegradation point of view, the optimal V:Ti molar ratio was 0.30, which degraded 62.5% of methylene blue after 210 min under daylight irradiation [83]. Imran et al. [84] synthesized sol–gel TiO_2_ NPs doped with Fe, Co, and S, and tested their capacity to degrade Congo red dye. The dopant molar ratio varied from 0.5% to 1.5% of Co. Titanium isopropoxide was the TiO_2_ precursor, ferric nitrate the iron, cobalt nitrate the cobalt, and thiourea the sulfur. After precipitation, the NPs were left in the solution for 100 °C and calcinated at 500 °C for 3 h. The catalyst degraded 99.3% of the dye in 1 h12″, showing a successful industrial and environmental applicability. The maximum absorbance was reported at 550 nm [84].

Jaimy et al. [85] attempted to achieve visible-light photocatalytic activity without the use of dopants. The authors studied the crystallization process of sol–gel anatase NPs (assisted by microwave treatment), where titanium (IV) oxysulphide was the precursor for the hydrolysis performed under basic conditions. The sol was microwave-treated (420 W) from 0 to 60 min. Methylene blue was used to prove the photodegradation activity in daylight. A shift in the titania workable wavelength into the visible range was observed. Moreover, microwave assistance improved the TiO_2_ crystallinity. A total of 94% of methylene blue was degraded under visible light [85].

The sol–gel nonhydrolytic route was also studied, in this case by Albrbar et al. [86] to produce TiO_2_ NPs doped with N and/or S. Cyclohexane and dimethyl sulfoxide were used as the solvents, and gelation temperatures of 160 °C and 200 °C were tested. N was added during the annealing phase (500 °C for 3 h, with NH_3_). The pollutant tested was the dye C.I. Reactive Orange 16. The best results for its degradation were obtained with S-N-doped TiO_2_ (it was able to completely degrade the pollutant in 30 min under visible light).

### 2.3. Amorphous Titania

Among academics, amorphous TiO_2_ has recently been a topic of tremendous interest. It has been shown to have a wide range of applications, for example as a high-performance photocatalyst [45,46], as a dye sensitizer [47] or electrode [48] in solar batteries, as a thin film in capacitors [46], in resistive random access memory applications [44], as a self-cleaning agent due to its super-hydrophilicity [45,46,87], or to purify dye-polluted water [88]. It has recently been put to work as anodes for sodium ion rechargeable batteries [89], low-temperature oxygen sensors [90], or visible-light photocatalysts [45,91].

The minimization of the recombination and rapping processes (along with diffusional distances) is another way to optimize titania photocatalysis performance. Here, amorphous thin films and NPs offer excellent potential. The absence of phase boundaries in amorphous materials gives a great contribution to reducing rapping processes. Until now, the study of TiO_2_ photocatalytic properties has been centered on titania crystalline polymorphs. Nevertheless, amorphous titania (a-TiO_2_) is often the first phase to form in many wet chemical synthetic routes (the first nuclei to form are anatases that rapidly evolve to oxy-hydroxy amorphous phases [44,45,46,87]). Amorphous TiO_2_ is easier to process into different forms and its lower-order state (with a greater number of structural defects) allows a much wider range with higher levels of dopants. Furthermore, the much easier oxygen removal from amorphous titania than from crystalline titania contributes to a larger number of surface reaction sites in amorphous titania.

Seo et al. [92] sol–gel-synthesized amorphous peroxy-titania to be used in the treatment and removal of organic pollutants. Titanium (IV) isopropoxide and hydrogen peroxide were mixed under soft conditions (powder drying at 50 °C for 4 h). The organic pollutant used for the photodegradation assays was 4-chlorophenol. After 4 h under visible light, the amorphous peroxy-titania could degrade 96% of the 4-chlorophenol, whereas other photocatalysts reached a maximum of 24% in the same period. When tested against other pollutants (phenol, acetaminophen, benzoic acid, and carbamazepine), it fully degraded acetaminophen and phenol, although it barely degraded benzoic acid and carbamazepine [92].

In 2019, Lee et al. [93] studied the photocatalytic activity of the amorphous Ti-peroxy complex. This composite was synthesized by mixing hydrogen peroxide with titanium hydride, to first form a shell of TiOH over the TIH_2_ core. Then, H_2_O_2_ was added to the solution to form amorphous TiO_2_ with surface peroxide groups (a yellow gel was obtained by adding 200 mL of H_2_O_2_, and green one was obtained by adding 150 mL of H_2_O_2_). The product was dried at 100 °C. To test the photocatalytic activity, rhodamine B (RhB) was used. Both the green and yellow photocatalysts were able to degrade RhB up to 95%. Table 2 summarizes the research works presented in this short review.

Dorosheva et al. [94,95] recently studied the influence of different annealing temperatures on the formation of TiO_2_ nanoparticles using a sol–gel method. In a first attempt, the material was annealed for one hour at a temperature varying from 200 to 1000 °C [94], while in a different study, the nanoparticles were also treated at 800 °C starting at 30 min until 240 [95]; in both cases, the annealing was carried out under a nitrogen atmosphere. The results showed a difference in the crystal structure of the nanoparticles where anatase was observed at 400 and 600 °C, while a, X-ray diffraction (XRD) rutile pattern was present at 800 and 1000 °C. Beyond that, the band gap of the NPs decreased within the increment in the annealing temperature, reaching the lowest value of 2.6 eV at 1000 °C [94]. However, the effect of incrementing the time of annealing at 800 °C could only influence the rise in crystallinity and oxygen vacancy, demonstrating no influence over the crystal structure or the band gap [95].

Chung et al. [96] produced amorphous titania with a sol–gel process and doped it with nitrogen, using heat treatment at 130 °C. The amorphous N/TiO_2_ after being calcinated (at 500 °C) transformed into anatase N/TiO_2_. Both materials showed a band gap at 2.4 eV. The photocatalytic activity was tested against formaldehyde and methylene blue under visible-light irradiation, obtaining a degradation of 60% of formaldehyde and 30% of methylene blue after 2 h of reaction time [96].

A recent review reported recent developments in the heterostructures of Ti and Sn oxides used as photocatalysts. The revised techniques are hydrothermal, sol–gel, electrospinning, precipitation, and the combination of these methods for the production of undoped, doped, or ternary TiO_2_-SnO_2_ materials and their use for the degradation of organic pollutants from water [97].

**Table 2 materials-16-02731-t002:** Visible-light degradation yield of different materials.

Ref.	Strategy	Best Composition	Morphology	PC Testing	Best Activity	Crystal Structure
[98]	D	Au–CuS–TiO_2_	Nanoparticles	OTC	68%	Anatase (tetragonal)
[75]	C	TiO_2_-Au-CeO_2_	Hollow sphere	MO	95%	Anatase (tetragonal)
[99]	C	Mn_3_O_4_(ZnO/TiO_2_)	Nanoparticle (layered)	DNT	71%	Amorphous
[74]	C	TiO_2_-PDA NP	Nanoparticles/coating	GS; FR	90% / 98%	Rutile (tetragonal)
[70]	C	2 Au-(6TiO_2_/CN)	Nanoparticles/nanosheets	2,4-DCP; BPA	46% / 37%	Anatase (tetragonal)

Legend: PCN-222—metalloporphyrinic metal–organic framework; PDA—polydopamine; GO—graphene oxide; PC—photocatalysis; D—doped; C—composite; OTC—oxytetracycline; DFC—diclofenac; MB—methylene blue; PH—phenol; BPA—bisphenol A; RhB—rhodamine B; 2,4-DCP—2,4-dichlorophenol; 4-chlorophenol—4-CP; RE—real effluent; MO—methyl orange; OFC—ofloxacin; DNT—denitrification; GS—geosmin; FR—fluorine.

## 3. Discussion

In general, semiconductors, in particular TiO_2_, are a class of materials that have been the subject of several efforts to overcome their limitations, increase their efficiency, and broaden their potential applications.

The nanocomposite strategy produces the most promising environmental remediation results. Numerous studies [71,75,78] that virtually completely degraded environmental contaminants attest to this fact. Furthermore, it should be noted that only Herrera et al. (2020) [79] worked on a real-world context, where their titania-based nanoparticles only degraded 40% of the environmental contaminants when exposed to visible light. The recovery of the photocatalyst after its use, which is one of the primary goals in environmental remediation, is another critical vector still absent from most research works. Zhao et al. (2016) [73] presented a novel idea by using magnetic nanoparticles that can be easily recovered (after use) with a simple magnet.

The catalyst Au-(T/CN) nanocomposites have high and stable photodegradation activities in water for the photocatalytic degradation of 2,4-DCP and BPA under visible-light irradiation. However, less than 50% of the tested pollutants were degraded (46 and 37 percent, respectively). This finding is an improvement when compared to the results of bare g-C_3_N_4_ and commercial P-25, suggesting that SPR-assisted photocatalysts can achieve the rapid and efficient degradation of pollutants. This behavior is attributed to the nanocomposites’ extended visible-light absorption at 590 nm, which promotes charge separation by transferring electrons from Au and g-C_3_N_4_ to TiO_2_. Moreover, the use of Au nanoparticles to photosensitize nanocomposites could have a significant impact on the creation of highly effective visible-light sources in the near future, although higher concentrations of NPs may lead to a decrease in their photocatalytic activity [70].

Core–shell nanostructures: ZnO/TiO_2_ shell and Mn_3_O_4_ cores were used to create multi-interface type II heterojunctions, which aided in the separation of charge carriers and the production of active radicals. The structure’s logical design produced excellent results for denitrification from water, with a NO_3_^−^-N reduction efficiency of 98.6% and a N_2_ selectivity of 98.8% under concurrent visible light and ultrasound irradiation for 120 min. The controlled fabrication of multi-interface heterojunctions using atomic layer deposition was a promising method to obtain a good-quality material. Additionally, the photocatalytic system powered by visible light and ultrasonic assistance offered promising potential for selective denitrification from aquatic environments [99].

The TiO_2_/PDA core–shell nanoparticles and the TiO_2_/PDA/GA coating showed dramatically increased absorption, not only in the visible region but also in the UV region when compared to pure TiO_2_. Additionally, it should be highlighted that the absorption edges of the core–shell nanoparticles and the coating clearly shifted toward the red in comparison to TiO_2_. The visible-light response was improved with the immobilization of a PDA shell onto the TiO_2_ core, which reduced the band gap energy of the TiO_2_ nanoparticles. To avoid separation processes in the continuous application of the modified materials, secondary modification by GA must be conducted. PDA and GA modifications successfully promote the produced rutile TiO_2_’s photocatalytic capacity when exposed to visible light. One of the main obstacles preventing the widespread use of nanoparticles is separation. Circulating techniques are challenging to implement in addition to requiring separation and filtering because dispersed nanoparticles create particulate suspensions. Therefore, in this situation, a TiO_2_/PDA/GA coating was self-assembled to address the drawbacks of the direct use of nanoparticles by utilizing the cross-linking action of GA and PDA [74].

## 4. Conclusions

This short review presents recent work on the environmental applications of daylight-photoactive titania nanoparticles. Titania composite, titania doping, and more recently, titania amorphous structures—three distinct techniques—all indicated interesting potential.

Doping and composite titania NPs appear to have made significant contributions to the environmental field. However, despite being the main objective of wastewater treatment plants, there are still not enough industrial applications. The focus of current research is on the recovery of photocatalyst materials as well as their stability and longevity.

Promising results were shown for amorphous TiO_2_ NPs which can help to overcome some of the observed limitations of crystalline TiO_2_ applications. For all the presented studies, either for the use of sensitizers for SDT or as a bactericidal agent, amorphous TiO_2_ NPs do not need UV light stimulation, and the synthesis process to obtain these amorphous structures is considerably easier and more immediate. Nevertheless, there is still much work to carry out related to the scale-up of wastewater treatment plants. To first test the materials at a controllable and smaller level, but keeping this in a real scenario, could be difficult since there are other chemicals and pollutants involved which may inhibit the photocatalysts’ performance. Researchers interested in this field can take advantage of the many photocatalyst options listed in this review and start to collaborate and think of strategies to release these materials onto the market to reach, when feasible, wastewater treatment plants. 

## Figures and Tables

**Figure 1 materials-16-02731-f001:**
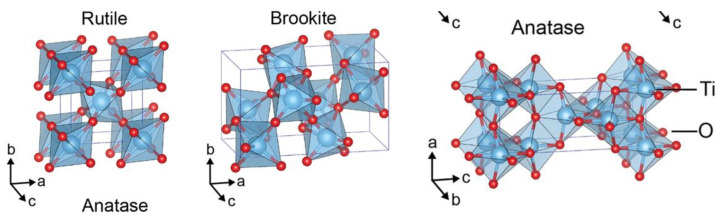
Titania polymorphs structures: rutile, brookite, and anatase (CC BY 4.0 [37]).

**Figure 2 materials-16-02731-f002:**
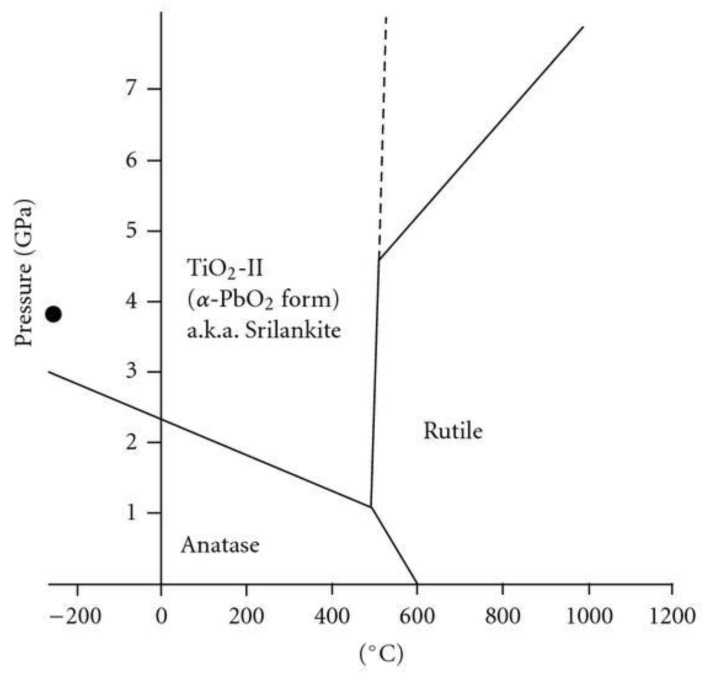
Titania phase diagram (CC BY 3.0 [41]).

**Figure 3 materials-16-02731-f003:**
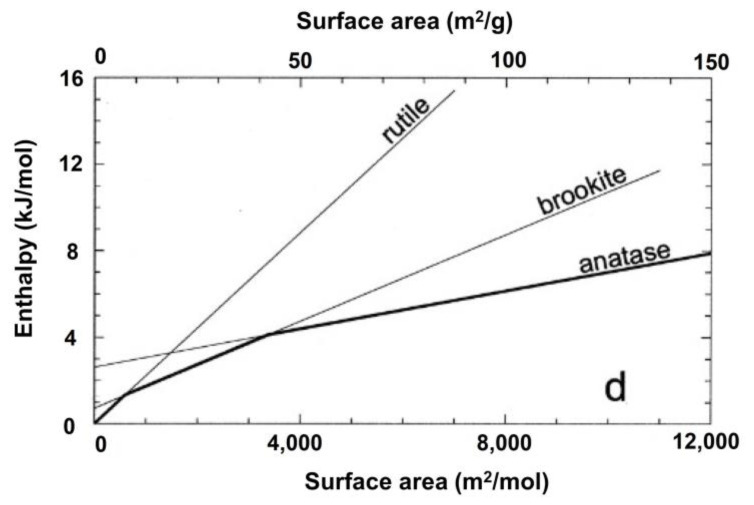
Titania polymorphs stability at nanoscale [42] Copyright (2002) National Academy of Sciences, Washington, WA, USA.

**Figure 4 materials-16-02731-f004:**
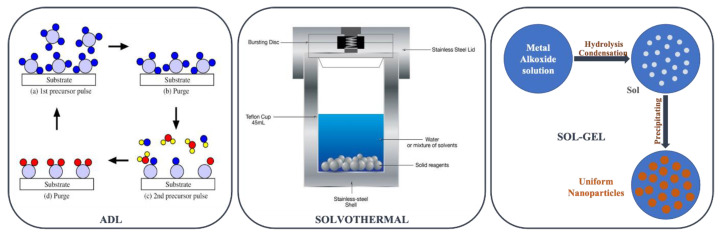
ALD, solvothermal, and sol–gel titania methodologies.

**Figure 5 materials-16-02731-f005:**
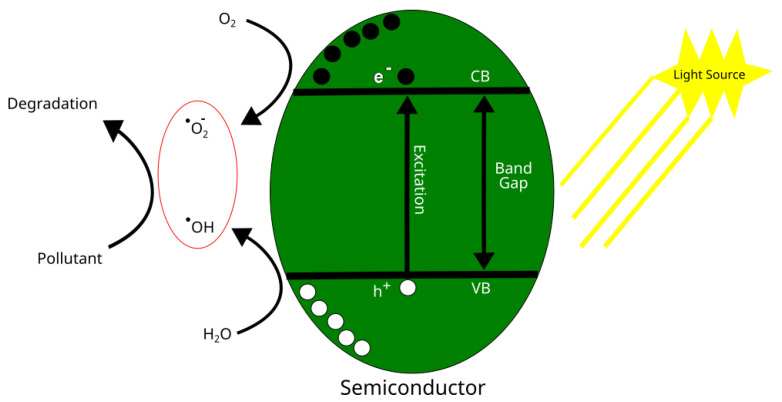
Schematic of photocatalysis process applied to environmental remediation.

**Table 1 materials-16-02731-t001:** Titania polymorphs’ properties [35,36].

Anatase	Crystal System	Tetragonal
	Unit Cell	a = 3.7845; c = 9.5143 (Å)
	Space Group	I4_1_/amd
	Band Gap (eV)	3.23–3.59 (Theoric)
Brookite	Crystal System	Orthorhombic
	Unit Cell	a = 5.4558; b = 9.1819; c = 5.1429 (Å)
	Space Group	Pbca
	Band Gap (eV)	~3
Rutile	Crystal System	Tetragonal
	Unit Cell	a = 4.5937; c = 2.9587 (Å)
	Space Group	P4_2_/mnm
	Band Gap (eV)	3.02–3.24 (Theoric)

## Data Availability

Not applicable.

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
