# Peer review of "Daylight Photoactive TiO2 Sol-Gel Nanoparticles: Sustainable Environmental Contribution"

_materials, 2023, doi:10.3390/ma16072731_

Round 1

Reviewer 1 Report

The present review manuscript entitled “Daylight photoactive TiO2 sol-gel nanoparticles: sustainable environmental contribution” authored by Daniel Barcelos describes the sol-gel titania-based nanoparticles with enhanced visible light photo-activity in the environmental area. Furthermore, titania co-doping, titania composite design, and recently amorphous networks have been the most used approaches to report this subject. The authors report an interesting review manuscript approach and the presentation of the work is clear. The objective and justification of the work are clear. I would recommend it for publication in Materials. However, certain Minor issues are detailed below to improve the quality of the manuscript.

I advise the authors to take the following points into account while revising their manuscript.

Comment 1: Add the list of acronyms/ abbreviations in the revised manuscript.

Comment 2: There are some typographical and subscript errors in the manuscript text, so the authors need to correct them in the revised manuscript and also improve the English language of the manuscript.

Comment 3: Abstract should discuss the problem statement in the manuscript text. So, revise the abstract section.

Comment 4: Provided Graphical Abstract is looking fine. However, if authors can improve it to some more extent, will attain a broad readership. So try to rearrange the graphical abstract.

Comment 5: More recent references from the year 2022 are required in the revised manuscript. So, please include the below-mentioned references and discuss them in the revised manuscript.

Nachit, W., H. Ait Ahsaine, Z. Ramzi, S. Touhtouh, I. Goncharova, and K. Benkhouja. "Photocatalytic activity of anatase-brookite TiO2 nanoparticles synthesized by sol gel method at low temperature." Optical Materials 129 (2022): 112256.

Rajput, Rekha B., Shweta N. Jamble, and Rohidas B. Kale. "A review on TiO2/SnO2 heterostructures as a photocatalyst for the degradation of dyes and organic pollutants." Journal of Environmental Management 307 (2022): 114533.

Bachvarova-Nedelcheva, Albena, Reni Iordanova, Anton Naydenov, Angelina Stoyanova, Nelly Georgieva, Veronica Nemska, and Tsvetelina Foteva. "Sol-Gel Obtaining of TiO2/TeO2 Nanopowders with Biocidal and Environmental Applications." Catalysts 13, no. 2 (2023): 257.

Comment 6: Figures 3-5 resolution is very poor, so improve the quality of the figures.

Comment 7: Include some information regarding other synthesis approaches of TiO2 nanoparticles and discuss them in the revised manuscript and highlight the significance of the sol-gel approach in comparison with other synthesis approaches.

Comment 8: Include and discuss the factors affecting TiO2 nanoparticle synthesis.

Comment 9: A more detailed and critical literature review discussion is required for all the sub-sections with their detailed pros and cons.

Comment 10: Please check whether the authors have already obtained the copyright permissions for Figures 4-7 used in the manuscript.

Comment 12: In Section 3, the Discussion needs to be elaborated with some more references.

Comment 13: Revise the conclusion section with future perspectives.

Comment 14: The homogeneity of the reference section needs to be maintained. In some references, journal names are written in full form and some in abbreviation form, and also subscript errors. So please check and revise accordingly to the journal instructions.

Author Response

Reviewer #1
General comments:

We thank the reviewer for the general overview of our manuscript for thoroughly reading the document and specifying the different points in the specific comments. All of these have been addressed and we believe that the changes have enhanced the quality of our work.

The present review manuscript entitled “Daylight photoactive TiO2 sol-gel nanoparticles: sustainable environmental contribution” authored by Daniel Barcelos describes the sol-gel titania-based nanoparticles with enhanced visible light photo-activity in the environmental area. Furthermore, titania co-doping, titania composite design, and recently amorphous networks have been the most used approaches to report this subject. The authors report an interesting review manuscript approach and the presentation of the work is clear. The objective and justification of the work are clear. I would recommend it for publication in Materials. However, certain Minor issues are detailed below to improve the quality of the manuscript.

I advise the authors to take the following points into account while revising their manuscript.

Comment 1: Add the list of acronyms/ abbreviations in the revised manuscript.

A list of acronyms/abbreviations was added to the manuscript, thank you for the comment.

List of acronyms/abbreviations

Nanoparticles - NPs

Food and Drug Administration - FDA

Ultraviolet - UV

Acrylonitrile butadiene styrene - ABS

Polyethylene terephthalate - PET

Polybutylene terephthalate - PBT

Polyvinyl chloride - PVC

Advanced oxidation processes - AOPs

Atomic layer deposition - ALD

Valence band - VB

Conduction band - CB

Reactive oxygen species - ROS

Graphene oxide - GO

Dichlorophenol - DCP

Polystyrene - PS

Cetyltrimethyl ammonium bromide - CTAB

Metal organic frameworks - MOFs

Rhodamine B - RhB

X-ray diffraction - XRD

Metalloporphyrinic metal-organic framework - PCN-222

Polydopamine - PDA

Photocatalysis - PC

Doped - D

Composite - C

Oxytetracycline - OTC

Diclofenac - DFC

Methylene blue - MB

Phenol - PH

Bisphenol A - BPA

2,4-dichlorophenol - 2,4-DCP

4-chlorophenol - 4-CP

RE – Real Effluent - RE

Methyl orange - MO

Ofloxacin - OFC

Denitrification - DNT

Geosmin - GS

Fluorine - FR

Comment 2: There are some typographical and subscript errors in the manuscript text, so the authors need to correct them in the revised manuscript and also improve the English language of the manuscript.

OK; done. Corrections in the manuscript and English was revised. Thanks for the comment.

Comment 3: Abstract should discuss the problem statement in the manuscript text. So, revise the abstract section.

A new abstract was written to suit the manuscript, thank you for the comment.

Visible light photo-active titania nanomaterials excelled recently in wide range of industrial areas, particularly in environmental remediation. Sol-gel methodology, by another hand, has been successfully used to synthesize either crystalline and amorphous TiO2 micro- and nanoparticles, due to its outstanding chemical simplicity and versatility, along with extensive spectrum of possible product morphologies. This short review aims to bring and discuss the most recent developments in visible light photo-active titania-based nanoparticles, in the environmental remediation area. Titania co-doping, titania composite design and recently amorphous networks have been the most used strategies to address this goal. Finally, a prospective regarding the future of these fields is given.’

Comment 4: Provided Graphical Abstract is looking fine. However, if authors can improve it to some more extent, will attain a broad readership. So try to rearrange the graphical abstract.

A new graphic abstract is presented.

Comment 5: More recent references from the year 2022 are required in the revised manuscript. So, please include the below-mentioned references and discuss them in the revised manuscript.

Nachit, W., H. Ait Ahsaine, Z. Ramzi, S. Touhtouh, I. Goncharova, and K. Benkhouja. "Photocatalytic activity of anatase-brookite TiO2 nanoparticles synthesized by sol gel method at low temperature." Optical Materials 129 (2022): 112256.

Bachvarova-Nedelcheva, Albena, Reni Iordanova, Anton Naydenov, Angelina Stoyanova, Nelly Georgieva, Veronica Nemska, and Tsvetelina Foteva. "Sol-Gel Obtaining of TiO2/TeO2 Nanopowders with Biocidal and Environmental Applications." Catalysts 13, no. 2 (2023): 257.

The articles above indicated cannot be added to the article review since it does not present a photocatalytic activity under visible-light which is the center topic of the review.

Rajput, Rekha B., Shweta N. Jamble, and Rohidas B. Kale. "A review on TiO2/SnO2 heterostructures as a photocatalyst for the degradation of dyes and organic pollutants." Journal of Environmental Management 307 (2022): 114533.

The indicated articles were added and discussed in the manuscript, thank you for the indications.

Comment 6: Figures 3-5 resolution is very poor, so improve the quality of the figures.

Figures were altered to improve the quality of them. Thanks for the comment.

Comment 7: Include some information regarding other synthesis approaches of TiO2 nanoparticles and discuss them in the revised manuscript and highlight the significance of the sol-gel approach in comparison with other synthesis approaches.

New paragraphs about the synthesis were added to the manuscript.

Comment 8: Include and discuss the factors affecting TiO2 nanoparticle synthesis.

Some paragraphs have been added.

Comment 9: A more detailed and critical literature review discussion is required for all the sub-sections with their detailed pros and cons.

A more organized approach has been performed.

Comment 10: Please check whether the authors have already obtained the copyright permissions for Figures 4-7 used in the manuscript.

Figures 7 was made by the authors so no copyright is required. Others figures are based on (not copies).

Comment 12: In Section 3, the Discussion needs to be elaborated with some more references.

OK, done.

Comment 13: Revise the conclusion section with future perspectives.

OK, done.

Comment 14: The homogeneity of the reference section needs to be maintained. In some references, journal names are written in full form and some in abbreviation form, and also subscript errors. So please check and revise accordingly to the journal instructions.

Alterations and a review of the references were made, thank you for the comment.

Reviewer 2 Report

This is a very interesting review article that probably can be recommended for publication, but only after clarifying and detailing some parts of the text.

1.     Titania Polymorphs. It would be extremely helpful if the authors showed in Table form a comparison of the main properties of different polymorphs (lattice parameters, band gap et).

2.     Page 6. Sentence “Synthetic titania (micro- or NPs) is, laboratory or industrially, produced through atomic layer deposition (ALD), hydrothermal, solvothermal or sol-gel methodologies …”

This sentence needs supporting references for each method. Note that recently, others methods, such extraction-pyrolytic method, have been successfully used:

Serga, Vera, et al. "Study of phase composition, photocatalytic activity, and photoluminescence of TiO2 with Eu additive produced by the extraction-pyrolytic method." Journal of materials research and technology 13 (2021): 2350-2360. https://doi.org/10.1016/j.jmrt.2021.06.029

3.     I would like to draw your attention to several new works on sol-gel nano-TiO2:

Dorosheva, I.B., Valeeva, A.A., Rempel, A.A. et al. Synthesis and Physicochemical Properties of Nanostructured TiO2 with Enhanced Photocatalytic Activity. Inorg Mater 57, 503–510 (2021). https://doi.org/10.1134/S0020168521050022

B. Dorosheva, E. V. Adiyak, A. A. Valeeva, and A. A. Rempel. Synthesis of nonstoichiometric titanium dioxide in the hydrogen flow. AIP Conference Proceedings 2174, 020019 (2019); https://doi.org/10.1063/1.5134170

4. I would also like to see an additional paragraph or two on the usefulness of ab initio calculations for the problems at hand.

5. Table 1. For each line it would be useful to see the crystalline structure of TiO2. It is important to note that there are several modification of TiO2 and they give different effects. See, some of them:  Tsebriienko, T.; Popov, A.I. Effect of Poly(Titanium Oxide) on the Viscoelastic and Thermophysical Properties of Interpenetrating Polymer Networks. Crystals 202111, 794. https://doi.org/10.3390/cryst11070794

In general, the manuscript is interesting and can be recommended for publication after constructive reflection on the above comments.

Author Response

Reviewer #2

We thank the reviewer for the general overview of our manuscript for thoroughly reading the document and specifying the different points in the specific comments. All of these have been addressed and we believe that the changes have enhanced the quality of our work.

General comments:

This is a very interesting review article that probably can be recommended for publication, but only after clarifying and detailing some parts of the text.

  1. Titania Polymorphs. It would be extremely helpful if the authors showed in Table form a comparison of the main properties of different polymorphs (lattice parameters, band gap et).

A table with the properties of titania polymorphs was added to the manuscript. Thank you for the indication.

Anatase

Crystal System

Tetragonal

Unit Cell

a = 3.7845 ; c = 9.5143 (Å)

Space Group

I41/amd

Band Gap (eV)

3.23-3.59 (Theoric)

Brookite

Crystal System

Orthorhombic

Unit Cell

a = 5.4558; b = 9.1819; c = 5.1429 (Å)

Space Group

Pbca

Band Gap (eV)

~3

Rutile

Crystal System

Tetragonal

Unit Cell

a = 4.5937; c = 2.9587 (Å)

Space Group

P42/mnm

Band Gap (eV)

3.02-3.24 (Theoric)

  1. Page 6. Sentence “Synthetic titania (micro- or NPs) is, laboratory or industrially, produced through atomic layer deposition (ALD), hydrothermal, solvothermal or sol-gel methodologies …”This sentence needs supporting references for each method. Note that recently, others methods, such extraction-pyrolytic method, have been successfully used:

Serga, Vera, et al. "Study of phase composition, photocatalytic activity, and photoluminescence of TiO2 with Eu additive produced by the extraction-pyrolytic method." Journal of materials research and technology 13 (2021): 2350-2360. https://doi.org/10.1016/j.jmrt.2021.06.029

New references were added to these as examples of successful production of titania nanoparticles.

  1. I would like to draw your attention to several new works on sol-gel nano-TiO2:

Dorosheva, I.B., Valeeva, A.A., Rempel, A.A. et al. Synthesis and Physicochemical Properties of Nanostructured TiO2 with Enhanced Photocatalytic Activity. Inorg Mater 57, 503–510 (2021). https://doi.org/10.1134/S0020168521050022

  1. Dorosheva, E. V. Adiyak, A. A. Valeeva, and A. A. Rempel. Synthesis of nonstoichiometric titanium dioxide in the hydrogen flow. AIP Conference Proceedings 2174, 020019 (2019); https://doi.org/10.1063/1.5134170

Thank you for the indication of these interest works they were added to the review.

  1. I would also like to see an additional paragraph or two on the usefulness of ab initio calculations for the problems at hand.

OK, done

With reference to electronic band gap in crystalline titania, it is well established, both theoretically and experimentally. Nevertheless the understanding of the structural and electronic properties of amorphous titania phases (a-TiO2) is on the agenda with ab-initio calculations for a-TiO2, based on the similarity of shape of the electronic density of states in the conduction band of amorphous and crystalline TiO2 phases [Molecules 2018, 23(7), 1677; https://doi.org/10.3390/molecules23071677 ]

  1. Table 1. For each line it would be useful to see the crystalline structure of TiO2. It is important to note that there are several modification of TiO2 and they give different effects. See, some of them: Tsebriienko, T.; Popov, A.I. Effect of Poly(Titanium Oxide) on the Viscoelastic and Thermophysical Properties of Interpenetrating Polymer Networks. Crystals 2021, 11, 794. https://doi.org/10.3390/cryst11070794

The crystalline structure information was added in the table 2.

In general, the manuscript is interesting and can be recommended for publication after constructive reflection on the above comments.

Reviewer 3 Report

In this review, the authors focus on titania, a common photocatalyst material. The background information of titania is comprehensively introduced, followed by listing the reported strategies of enhancing photocatalytic activity of titania. I appreciate the authors’ effort. But I think this manuscript is not qualified for a review that can attract reader’s interest. In addition, the main text is strayed off the title. Thus, I could not recommend it publishing on Materials before further revision is made.

Comments:

1.     In the title, it is apparently the topic of current manuscript is about titania nanoparticles, especially those synthesized via sol-gel method. However, there are only two short paragraphs actually discussing about the nanoparticle-related stuff in the Introduction. Most of the introduction is talking about the basic background of titania in the natural world and industry in a Wikipedia style. I suggest the authors spending more time on the Introduction to make it more related to the topic because it is important to attract readers for a review article.

2.     In the literature review section (Section 2), the authors just simply list the related literatures one by one. I think that a qualified review article should have logic flow, in other words, story, to organize the literatures. A lazy listing is absolutely not acceptable.

Besides, the subsection index 1.1 is missing.

In my opinion, a review article should also have novelty in its topic in order to attract readers’ interest. I believe that it is the goal to every journal to improve the attraction to readers. It is apparently that the current manuscript focuses on the sol-gel nanoparticles of TiO2 according to the title as well as the main text. However, little part is devoted to talking about “sol-gel” and “nanoparticles”, where the novelty actually stands. In other words, most text in the Introduction has little novelty if it is only limited to the basic background of TiO2. My suggestion is that the authors should reconsider the topic or modify the title, and paying more effort in discussing the sol-gel technique. Actually, there are many interesting aspects to discuss for sol-gel technique. For example, it is worth talking about the effect of synthesis condition on the sol-gel-grown film properties.

  In addition, I think the Section 2 requires re-writing, because there is no logic through the passage. It is just mechanically piling the literatures, one paragraph by one paragraph. I think a qualified review article needs respective, not a simple literature list.

Author Response

Reviewer #3
We thank the reviewer for the general overview of our manuscript for thoroughly reading the document and specifying the different points in the specific comments. All of these have been addressed and we believe that the changes have enhanced the quality of our work.

General comment

In this review, the authors focus on titania, a common photocatalyst material. The background information of titania is comprehensively introduced, followed by listing the reported strategies of enhancing photocatalytic activity of titania. I appreciate the authors’ effort. But I think this manuscript is not qualified for a review that can attract reader’s interest. In addition, the main text is strayed off the title. Thus, I could not recommend it publishing on Materials before further revision is made.

Comments:

  1. In the title, it is apparently the topic of current manuscript is about titania nanoparticles, especially those synthesized via sol-gel method. However, there are only two short paragraphs actually discussing about the nanoparticle-related stuff in the Introduction. Most of the introduction is talking about the basic background of titania in the natural world and industry in a Wikipedia style. I suggest the authors spending more time on the Introduction to make it more related to the topic because it is important to attract readers for a review article.

The authors aim was to write a review article on titania micro- and nanoparticles in industrial world. Although simple, the integrated information (and some of the details given, namely about nanoparticles polymorph stability) are not so well known (besides both master and PhD students skip over the titania impact on industry over time).

Further to re-write all the Introduction section the authors do need more time. And it implies all new manuscript, with another structure/philosophy.

  1. In the literature review section (Section 2), the authors just simply list the related literatures one by one. I think that a qualified review article should have logic flow, in other words, story, to organize the literatures. A lazy listing is absolutely not acceptable.

The literature review section was re-organized by topics (in a historic logic flow) and some comments have been added to help reader follow the authors ideas.

 Besides, the subsection index 1.1 is missing.

Section numbers corrected.

In my opinion, a review article should also have novelty in its topic in order to attract readers’ interest. I believe that it is the goal to every journal to improve the attraction to readers. It is apparently that the current manuscript focuses on the sol-gel nanoparticles of TiO2 according to the title as well as the main text. However, little part is devoted to talking about “sol-gel” and “nanoparticles”, where the novelty actually stands. In other words, most text in the Introduction has little novelty if it is only limited to the basic background of TiO2. My suggestion is that the authors should reconsider the topic or modify the title, and paying more effort in discussing the sol-gel technique.

The authors aim was to compile in a review article the history of titania micro- and nanoparticles in industrial world. Although simple, the integrated information (and some of the details given, namely about nanoparticles polymorph stability) are not so well known (besides both master and PhD students skip over the titania impact on industry over time).

Actually, there are many interesting aspects to discuss for sol-gel technique. For example, it is worth talking about the effect of synthesis condition on the sol-gel-grown film properties.

In fact, sol-gel titania films/coatings have a long history (and synthesis conditions to be discussed, namely as the sol-gel-grown film properties) but sol-gel titania nanoparticles is a very recent topic. Regarding sol-gel synthesis novelty and details when titania nanoparticles are concerned, few has been consolidated so far.

  In addition, I think the Section 2 requires re-writing, because there is no logic through the passage. It is just mechanically piling the literatures, one paragraph by one paragraph. I think a qualified review article needs respective, not a simple literature list.

The discussion was re-organized by topics (some recent references were added) and comments have been integrated to help reader follow the authors ideas.

Round 2

Reviewer 3 Report

After reading the revised manuscript, I think the authors have addressed most of my concern. Thus, I would like to recommend the acceptance of the current manuscript.